# Improved prediction of hiking speeds using a data driven approach

**Andrew Wood** [1] *, **William Mackaness**[2], **T. Ian Simpson**[1], **J. Douglas Armstrong**[1]

**1** School of Informatics, University of Edinburgh, Edinburgh, United Kingdom, **2** School of Geoscience, University of Edinburgh, Edinburgh, United Kingdom

* andrew.wood@ed.ac.uk

**Data Availability Statement:** Data cannot be shared publicly because it was accessed under the terms of the Ordnance Survey Educational User Licence, and cannot be made directly available to third parties The data underlying the results

## Abstract

Hikers and hillwalkers typically use the gradient in the direction of travel (walking slope) as the main variable in established methods for predicting walking time (via the walking speed) along a route. Research into fell-running has suggested further variables which may improve speed algorithms in this context; the gradient of the terrain (hill slope) and the level of terrain obstruction. Recent improvements in data availability, as well as widespread use of GPS tracking now make it possible to explore these variables in a walking speed model at a sufficient scale to test statistical significance. We tested various established models used to predict walking speed against public GPS data from almost 88,000 km of UK walking / hiking tracks. Tracks were filtered to remove breaks and non-walking sections. A new generalised linear model (GLM) was then used to predict walking speeds. Key differences between the GLM and established rules were that the GLM considered the gradient of the terrain (hill slope) irrespective of walking slope, as well as the terrain type and level of terrain obstruction in off-road travel. All of these factors were shown to be highly significant, and this is supported by a lower root-mean-square-error compared to existing functions. We also observed an increase in RMSE between the GLM and established methods as hill slope increases, further supporting the importance of this variable.

## Introduction

Knowing how fast people are able to walk between locations is critical information in many situations. In hiking and hillwalking scenarios, this information is vital for safety reasons. If you are leaving in the morning for a hike then it is good practice to provide an estimated return time such that emergency services can be contacted if you get into difficulty and do not return [1]. An inaccurate estimate for how long a route will take could lead to unnecessary callouts, or delay a callout in a situation where every minute is important. Furthermore, in circumstances where a hiker has gone missing, an accurate measure of walking speed can help to restrict a potential search area around a last known location. Finally, when out on a hike there are situations where hikers may be deciding whether to follow a footpath, or take a more direct cross-country route. Accurate estimates of the walking speed and time for both scenarios are required to be able to select the optimal route.

presented in the study are available from the sources listed in S1 File. URLs to the original Hikr and OpenStreetMap data are provided in S1 File The elevation and Lidar data used throughout were accessed through Digimap (link in S1 File). The specific data sources and resolutions are listed. For each data source, all available UK data at the time of the study was requested (the dates each dataset were accessed are also provided in S1 File). The study can be replicated by others if the listed data is downloaded, and detailed steps to replicate the data processing steps are provided in S2 File. Further, the original code used to process the data is available on Github (link at the end of Methods section) The Hikr and OpenStreetmap data are publicly available The data obtained through Digimap was accessed under an Educational User License. This is not fully public, but is available freely to all educational users - this covers: "All activities that a fair-minded and reasonable person would agree falls within the spirit and intention of 'Educational Use'. Educational use at all levels – including schools, colleges, universities and research councils, (whether on site or remotely); Educational Use within an Elective Home Education." (full licence here: https://digimap.edina.ac.uk/help/copyright-and-licensing/os_eula/).

**Funding:** Author Andrew Wood was funded by the UK Engineering and Physical Sciences Research Council (grant EP/R513209/1), https://www.ukri.org/councils/epsrc/ The funders had no role in study design, data collection and analysis, decision to publish, or preparation of the manuscript.

**Competing interests:** The authors have declared that no competing interests exist.

There are a multitude of factors which can impact the walking speed and time predictions for a route [2], although these can generally be split into two categories [3, 4]. The first category covers the individual effects which depend on who precisely is undertaking the walk, and when they are doing it. These effects include group size (larger groups often walk slower), age or fitness of participants, and weather conditions, as well as the aim of the walk (afternoon stroll vs. specific hike). The second category covers the fixed effects which will affect all individuals who attempt the same route. These include how steep the terrain is and whether the route is paved, along a track or in wild country.

Most of the individual effects cannot be modelled without considerable prior knowledge about the person who is planning a route. Therefore, most existing hiking route planners calculate the walking speed solely based on the terrain, and this is presented as the average time (or time range) it takes to complete a hike. It is then left up to the individual to tune the predicted time for a hike given their knowledge about personal ability and circumstances.

Formulae of varying complexity have been proposed to estimate human walking speed and time along a projected path. A popular early method that is still widely used was put forward by Naismith [5] which calculates walking time under normal conditions as:

"*an hour for every three miles on the map, with an additional hour for every 2,000 feet of ascent.*"

This approximates to a walking speed of 5 km/h with 10 minutes added on for every 100 m of ascent. This was later adjusted by Aitken [6], who introduced a reduced base movement speed of 4 km/h on surfaces which are not paths or roads. Naismith's rule is still used today by Scout groups and other casual hikers due to the ease of calculating walking time by hand using a paper map. However, despite the widespread use, Naismith's rule does have a well-known limitation; namely that the predicted speed does not change when descending a hill, regardless of the gradient.

An alternative hiking function proposed by Tobler [7], has become more popular in recent research and other situations where speeds do not need to be calculated by hand:

$$W = 6*exp(-3.5|S + 0.05|),$$

where
W = velocity (km/h)
S = gradient of slope.

Like Naismith's rule, this gives a speed of 5 km/h on flat ground, with a maximum speed of 6 km/h on a mild descent (around 3 degrees). In a similar manner to Aitken's correction, a factor of 0.6 is applied to the calculated speed for all off-road travel. Tobler's function avoids the issues seen in Naismith's rule when descending slopes, but it predicts a sharp peak in walking speed on mild descents, which may be unrealistic. The formulae discussed here are directly compared in Fig 1.

Other studies have also looked at providing alternative methods to calculate walking speeds [9–11], but all continue to use walking slope as the main variable to determine walking speed (with various multiplicative factors applied for off-road travel).

When exploring speeds of fell-runners, Arnet [12] suggested that movement velocity may be dependent on three factors: obstruction (with different factors applied depending on the kind of obstruction), ascent in the run direction (walking slope) and slope of the terrain (hill slope). The actual values used in Arnet's calculations cannot be directly applied to walking speeds as they were based on orienteering championships where participants were running.

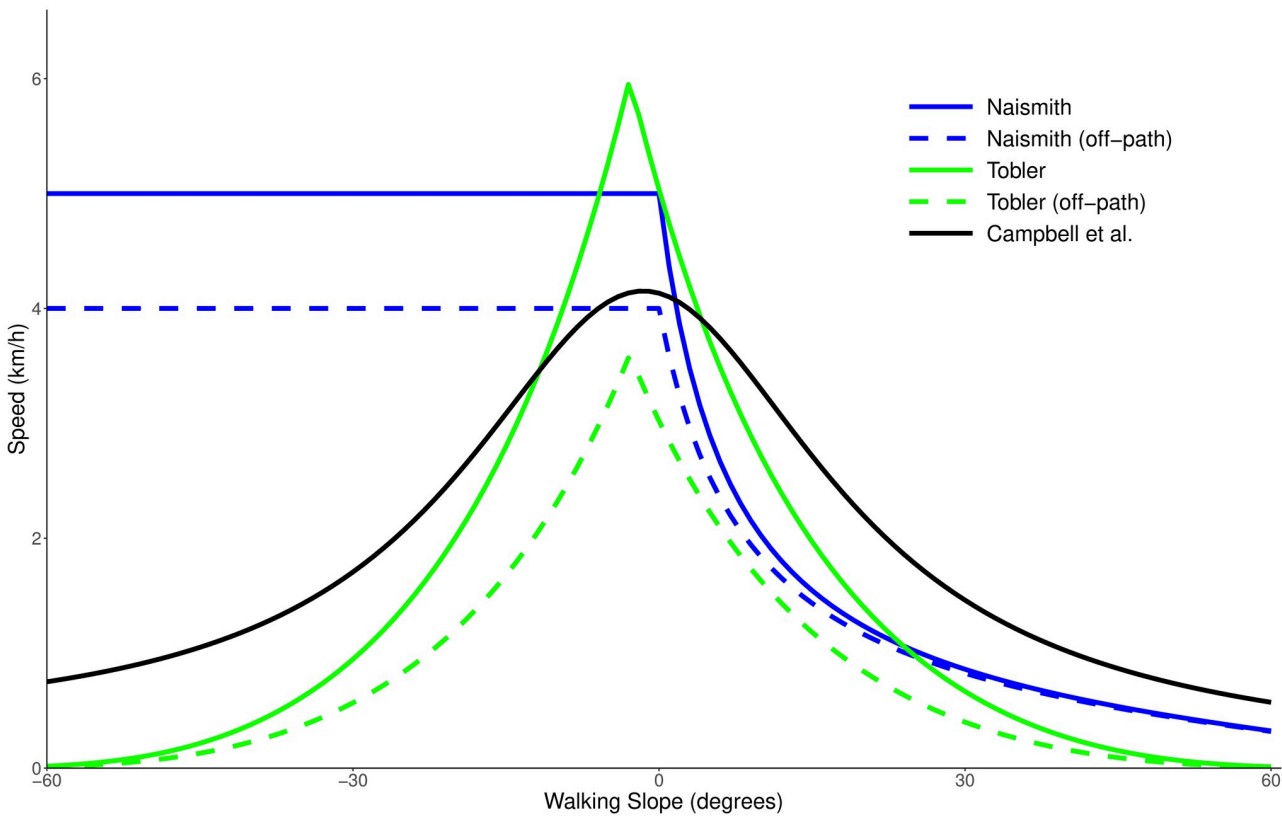

**Fig 1. Existing functions used to calculate walking speed.** Naismith's rule [5], Tobler's hiking function [7] and Campbell et al.'s function [8] plotted as predicted walking speed in km/h against the slope in the direction of travel (walking slope) in degrees where positive is uphill. For Naismith's function and Tobler's function, on and off-path versions are shown.

Experience tells us that traversing on a steep hill (while maintaining constant elevation) is more difficult than traversing flat ground. However, the existing methods estimate the same walking speed for both situations. Similarly, high levels of terrain obstruction in off-road areas (such as a thick gorse bush) are much more difficult to walk through than empty fields. The simple multipliers for off-road travel in Aitken's correction and Tobler's function do not provide any further distinction between two such regions.

Wood and Schmidtlein [13], took all three of Arnet's factors into account, and looked at evacuating citizens in the event of a hurricane. They applied Tobler's function to both the hill slopes and walking slopes, and calculated the terrain obstruction coefficients based on energy usage rather than walking speed (using [14]). They accepted that these were likely not the correct values, but were unable to find any better alternatives. Campbell, Dennison, and Butler [15] conducted a study using lidar data to explore the effects of ground roughness and vegetation density on firefighter evacuation speeds, but they did not consider the hill slope separately.

All of the studies mentioned above utilised relatively small sample sizes. However, the rise in use of global navigation satellite systems (GNSS), more frequently referred to as GPS tracking, means that a data-driven approach to modelling walking speed is now possible, which provides two main benefits. Firstly, it is possible to access GPS tracks from a wide variety of regions and terrains. Secondly, each track can easily be broken down into

individual sections, enabling specific route features to be investigated at much higher spatio-temporal resolution. This has been explored in recent work [8, 16], however the crowdsourced nature of these studies meant that data collection was not controlled, and thus that the data could not be assumed to consist wholly of walking or hiking tracks. In [16], data from hikes, jogs and runs was processed together, resulting in a very wide range of movement speed estimates. Campbell et al. attempted to overcome this in [8] by only considering data points with a speed between 0.2 m/s and 5 m/s (and the resulting model is shown in Fig 1). However, 5 m/s (18 km/h) is much higher than the maximum predicted speeds from existing methods (such as Naismith's rule), so it is likely some non-walking data remained. Furthermore, applying a blanket 0.2 m/s minimum speed may well overlook valid datapoints recorded by particularly slow individuals, or in especially difficult regions. Finally, although these studies had the benefit of using large sample sizes, they both looked solely at the effect of the walking slope on speed, and did not explore additional variables.

Here we used a data-driven approach to explore the impact of all three factors discussed by Arnet on walking speeds. These are the walking slope, the hill slope and the terrain obstruction. We aimed to use these factors to develop a model for the walking speed for an average individual. As with the existing methods, this model did not seek to model individual effects, and would still require tuning based on personal ability or conditions.

## Materials and methods

### Data set, cleaning and key assumptions

Full details of the various datasets used in this study are provided in S1 File. Further, a detailed description of the data filtering processes, and choices/assumptions made during data processing are described in S2 File.

In summary, GPS tracks were obtained for hikes in the UK from Hikr.org [17] and OpenStreetMap (OSM) [18]. Elevation and walking slope values were calculated and added to every GPS point using data from the Ordnance Survey Terrain 5 Digital Terrain Map (DTM), which provides elevation data at 5 m intervals across the whole of the UK [19]. Hill slope values were found using the quadratic surface method [20, 21]. Each data point was then classified as on a paved road, on an unpaved road, or off road, determined by searching a 50 m radius around each point in an OSM Road dataset [22]. Paved and unpaved road classification was determined using [23], with the unpaved road values being 'path', 'bridleway' and 'track'.

Terrain obstruction information was calculated using lidar datasets [24–26], as the difference in values between a Digital Surface Map (DSM) and Digital Terrain Map (DTM). This meant that any physical feature which protruded from the ground was regarded as an obstruction. We had access to lidar data at 2 m resolution covering large areas of England and Wales, but the coverage was not complete. Of our off-road data ($\sim$2,900 km, spread across over 1,200 tracks), over 2,000 km had lidar data available. Exploration of the lidar data (see S5 File) showed that there was a clear drop in walking speeds once the height of an obstruction was greater than 10 cm, beyond which the speed was relatively constant. We used this information to classify points into heavy obstruction ($>$10 cm) or light obstruction ($<$ = 10 cm) for modelling purposes.

Visual inspection of the tracks showed that a large number contained long breaks which could impact the accuracy of a walking speed model. Fig 2 shows examples of regions where breaks are visible in a GPS track, and the process developed to identify these regions is outlined in Algorithm 1.

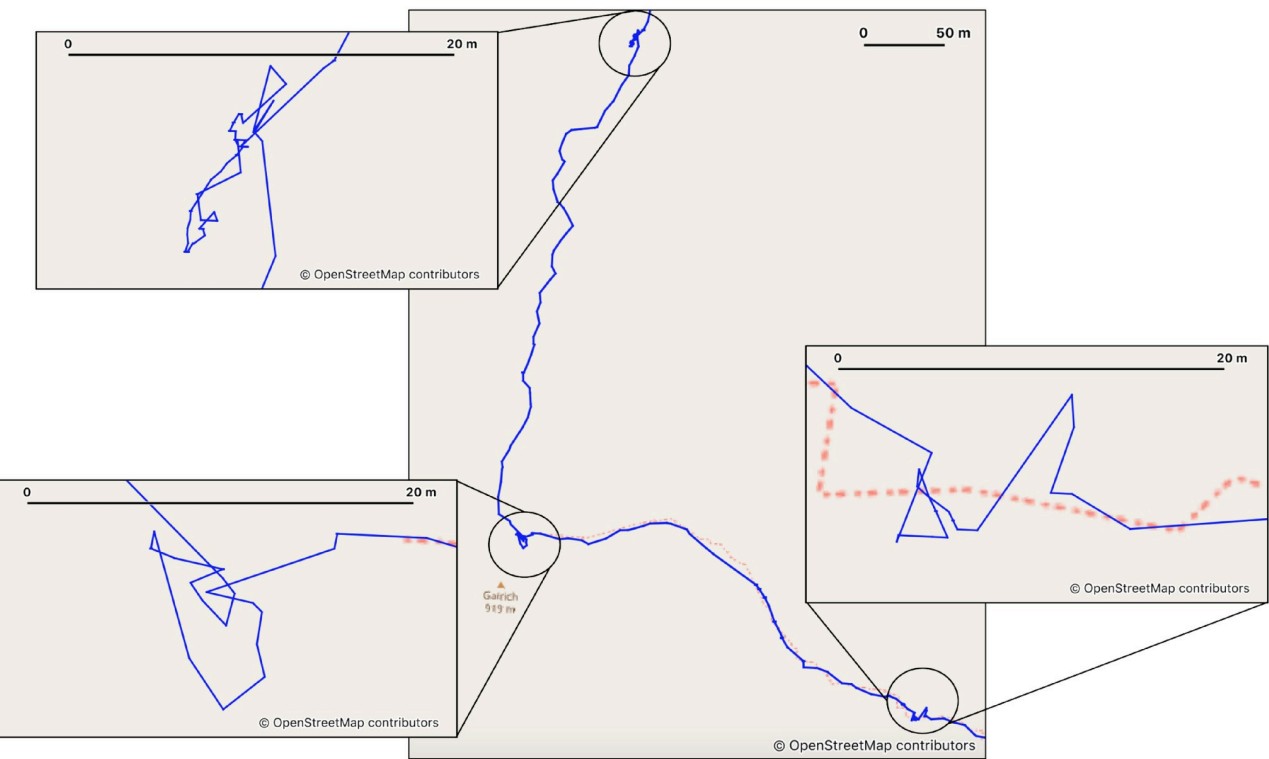

**Fig 2. A GPS track where 3 breaks can be identified by finding point clusters.** Clusters of points can form on a GPS track when a break is taken during a hike. By identifying these clusters as potential breaks we are able to remove most break periods from the datasets used for our analysis of walking speeds. For full details of these and other data filtering methods see S2 File. Background images from OpenStreetMap and OpenStreetMap Foundation [27], visualised using QGIS [28].

---

**Algorithm 1** Breakfinding process for a GPX track segment

```
1: Breakpoint_list = ∅
2: Find the median distance (r_median) and speed (s_median) of the segment
3: for point (p_i) in segment do
4:    Calculate travel direction quadrant and point angle
5:    Calculate break likelihood using the point speed and angle
6:    if speed == 0 or distance >1 km or duration >3 minutes then
7:      Breakpoint_list += p_i
8:    end if
9:    if speed >10 km/h and duration(p_{i-1}) >3 minutes then
10:      Breakpoint_list += p_i
11:    end if
12: end for
13: for point (p) in segment do
14:    if Neighbourhood of p is a cluster (C) then  ▷ See Defs 1 & 2,
          S2 File
15:      for point (p_c) in C do
16:        if Neighbourhood of p_c is a new cluster (C_n) then
17:          C = C ∩ C_n
18:        end if
19:      end for
20:      Remove points at the ends of the cluster with low break
          likelihood
```

```
21:     Add 'missing' points to the cluster (to make a continuous run
        of points) to form a Potential Break (B*)
22:     if less than half the points in B* have low break likelihood
        and there is travel in opposite quadrants (Q1 & 3 or Q2 & 4) then
23:        Breakpoint_list += B*
24:     end if
25:   end if
26: end for
```

Where the datapoints in the original GPS track were under 50 m in length, they were merged together to minimise the effects of errors in the GPS location values. While doing this, the resulting distance was the sum of all distances in the constituent GPS points, so may be longer than the straight line distance between co-ordinates. Similarly, both hill and walking slope values, as well as obstruction height, were calculated as the weighted average of constituent points, weighted by point duration.

While the Hikr dataset consisted of tracks which were tagged as a walk or hike, within some of these there were segments where it was clear that the participant was driving to or from the hike location, based on the observed speeds. The OSM data, on the other hand, was not filtered by transport type. There were a large number of tracks which were clearly from faster modes of transport, as their speed was implausible for a hiker. A process to remove these non-walking tracks and segments was created, whereby the known Hikr walking segments were used to create filtering bounds of plausible walking speeds, which could then be applied to the remainder of the dataset. This process is summarised in Algorithm 2.

**Algorithm 2:** Filtering process for GPS data from Hikr and OpenStreetMap

```
1: Remove duplicate segments (containing sections with identical start
location, end location, start time and duration)
2: Remove all segments with median speed >10 km/h
3: Remove all breaks with duration >30 seconds
4: Remove all breaks containing points with speed >10 km/h or distance
   >1 km
5: Merge remaining points into sections at least 50 m in length.
6: Recursively remove points with speed >10 km/h adjacent to a break,
   or the end of the track
7:
8: if Hikr data then
9:   if segment mean speed >10 km/h then
10:     remove segment
11:   end if
12:   Calculate filtering bounds        ▷ Eqs (1)−(4), S2 File
13: else
14:   Identify Key Points       ▷ see S2 File
15:   Remove single datapoints between Key Points
16:   Remove points where median speed between consecutive key points
   >Eq (1)
17:   while segment length is not consistent do
18:     Remove points with speed >10 km/h adjacent to a break, or the
       end of the track
19:     if segment median speed >Eq (1) or segment minimum speed >Eq
     (2) or segment upper quartile speed >Eq (3) or segment upper
       whisker speed <Eq (4) or segment duretion <2.5 minutes then
20:       Remove segment
21:     end if
22:   end while
23: end if
24:
25: Combine all segments into a single dataset
```

```
26: Remove the fastest and slowest 0.5% of the data
```

Following this, a decision was made to remove data from tracks found in Scotland. Lidar data covering the walking tracks was necessary to model the terrain obstruction, and was not sufficiently available in Scotland at the time of the study. Furthermore, analysis showed that that walking speeds in Scotland were at the extreme end of what is seen throughout the rest of the UK (see S4 File). Including this data without also including a corresponding extreme data-set where lidar data is available may result in incorrect modelling. All OSM track segments which took place within Scotland were excluded from further processing. Similarly Hikr tracks which were tagged as taking place in Scotland, and which fully took place in Scotland were excluded.

Our final modelling dataset consisted of 7,636 GPS tracks from England and Wales, with over 1.4 million individual data points and almost 88,000 km of travel. Each datapoint represented approximately 50–100 m of travel, and contained:

- Start coordinate

- End coordinate

- Start time

- Duration

- Distance

- Speed

- Elevation

- Walking slope

- Hill slope

- On-road flag

- Paved road flag (if on-road)

- Obstruction data available flag (if off-road)

- Heavy obstruction flag (if off-road and obstruction data available)

## Modelling

**Model formulation.**   Pilot studies were conducted to identify an appropriate model framework, using tracks within Scotland (see S3 File). Generalised linear model (GLM) and generalised additive model (GAM) approaches were explored, and within both we looked at the relationship between the walking and hill slopes, and the walking speed, with a small number of prior assumptions. As it is more challenging to walk on steeper slopes, for both the hill and walking slope components we knew that the walking speed should be a decreasing function of the magnitude of slope (with some allowance for faster walking speeds on mild descents). Models which failed to predict this were removed under the assumption that the data were overfitted. Furthermore, previous work [11, 29–31] has identified the existence of a critical gradient; the angle at which it is faster to zig-zag up a hill, rather than ascend directly. This occurs at a walking slope of around 15—21 degrees, so models which failed to predict the critical gradient occurring below 21 degrees were removed.

10-fold cross-validation was used to compare the remaining model parameters, looking at R-squared values, root-mean-squared error (RMSE) and mean absolute error. Where multiple models performed equally well, the simplest model was selected for ease of interpretabilty and real-world application. The selected model type was a Generalised Linear Model (GLM). Models were implemented using R version 3.6.1 [32].

**Terrain types.** Each of the three road types (paved road, unpaved road, off-road) was included in the model, both as factor variables, and as interaction terms with each of the slope variables.

Before adding terrain obstruction data to the model, we checked that there was no systematic difference between the walking speeds in regions where we had lidar data, and regions where we did not (see S5 File). Thus our findings in regions where lidar data was available could be extended to those where it was unavailable. Factor variables were then added to the model for each obstruction level (heavy, light or unknown obstruction).

**Statistical analysis.** Variables within the model were tested for significance using the Wald test, which allows us to account for correlation between points within the same track (`coeftest` function within `lmtest` package in R).

To measure the impact of our model, we compared walking speed predictions of our model against those of Naismith's, Tobler's and Campbell et al.'s models. Four different metrics were compared; the average percentage error, mean squared error (MSE), root-mean squared error (RMSE) and R squared value. These were explored when looking at both individual 50 m track sections, as well as predicted walking times for tracks as a whole. Finally, we isolated the off-road track sections in order to assess the improvement of our model at predicting walking speeds for off-path travel.

## Results

We started by assembling a dataset of hikes derived from approximately 20,000 public GPS tracks. These tracks recorded a variety of transport methods and required significant filtering. This process included iterative data cleaning to remove erroneous or non-walking data and identify/remove breaks (e.g. Fig 2) to give us a final usable dataset containing 7,636 GPS tracks, with over 1.4 million individual data points and covering almost 88,000 km of travel in the U. K. Each data point represents at least 50 m of travel (with a mean distance of 60.3 m), and the breakdown of the data by slope angle and terrain type is shown in Table 1. Previous research has found that most walking takes place on low walking slopes [33], and this is evidenced by our data ($\sim$98% of our data was from walking slopes of under 10 degrees).

Our curated hike dataset allowed us to create a data-driven model which we can directly compare with existing walking speed algorithms. The model formulation was selected using a small-scale exploratory study which considered data from Scotland (see S3 File). In this exploratory study, multiple different model types were explored which could fit the data, and which

**Table 1. Total distance of data under different terrain conditions (km).**

| | Hill Slope (degrees) | | | \|Walking Slope\| (degrees) | | |
|---|---|---|---|---|---|---|
| | **0–10** | **10–20** | **>20** | **0–10** | **10–20** | **>20** |
| Paved road | 62159.1 | 7841.2 | 2081.9 | 70726.5 | 1277.3 | 78.4 |
| Unpaved road | 9996.9 | 2210.3 | 700.7 | 12421.7 | 460.0 | 26.2 |
| Off Road (obstruction unknown) | 773.5 | 114.2 | 17.8 | 871.7 | 31.7 | 2.0 |
| Off Road (light obstruction) | 1282.9 | 150.1 | 23.8 | 1424.6 | 30.6 | 1.7 |
| Off Road (heavy obstruction) | 428.7 | 105.2 | 28.5 | 543.5 | 18.5 | 0.4 |

matched existing knowledge about walking speeds. Cross-validation methods showed that there was very little difference in performance of the best models, therefore the final model was a Generalised Linear Model (GLM), which was chosen as it was the simplest of those tested (we had no evidence that a more complex model would be superior). This choice also meant that our model was both easy to interpret, and simple to apply to future work.

This final GLM model included all three of the variables suggested by Arnet [12]:

$$v = exp(a + b\varphi + c\theta + d\theta^2) \tag{1}$$

where
$v$ = walking speed (km/h)
$\varphi$ = hill slope angle (degrees)
$\theta$ = walking slope angle (degrees)

Terrain obstruction level was included as a factor variable, while we considered the road types as both factor variables and interaction terms. Not all terms had a significant effect on all variables; we therefore created a model with all possible terms, and removed them one at a time (in order of least significance) until all remaining terms were significant to at least 95% confidence level (using Wald test). The final values for a, b, c and d are given in Table 2 for each of the terrain obstruction levels and road types. The critical gradient for this model is between 14—16 degrees when walking uphill and -16 – -18 degrees when walking downhill (depending on road and obstruction conditions), which is in line with previous findings.

Fig 3 shows the predicted walking speeds under different conditions. The importance of including both the hill slope and terrain obstruction variables can be clearly seen when looking at the Off Road Light Obstruction speed predictions. When directly ascending or descending a slope, the walking speed is comparable to walking on a road. However, when traversing a slope while off road, the walking speed is comparable to traversing a slope of double the gradient while on a road or path. Similarly, comparing the walking speed predictions of Off Road Light Obstruction and Off Road Heavy Obstruction reveals that just 10 cm of vegetation (our cutoff point for heavy obstruction) can reduce the walking speed by more than 0.5 km/h.

Fig 4 shows the same walking speed predictions as Fig 3, alongside the confidence interval for the mean walking speed for each terrain type. In the low-slope regions where most walking occurs, our model fits closely with the mean data confidence intervals. Our model does deviate from the confidence interval in some areas, particularly in high-slope and off-road regions. However, these are also the areas where we have the least amount of data (see Table 1). In Fig 4J the confidence interval for the mean would suggest that it is faster to walk on hill slopes of 30 degrees than hill slopes of 10 degrees. We have less than 30 km of data recorded in heavy obstruction regions on hill slopes of over 20 degrees, and less than 20 km of this had a walking slope magnitude of under 5 degrees (indicating that the slope was being traversed). Further, even within this range, the data is skewed towards the lower hill slope values. This lack of data explains the widening confidence interval, and counter-intuitive observations and we suggest that a targeted study would be required to collect more data in this region.

Fig 5 compares the Paved Road and Off Road Heavy Obstruction speed predictions from our model against the existing functions from Naismith, Tobler and Campbell et al. When looking at the walking slope, the largest areas of deviation between our model and Naismith's rule occurs when descending a slope, as Naismith's rule does not predict a reduced speed in this scenario. For both Tobler's and Campbell et al.'s functions, the shape of the walking slope component is relatively similar to our new model, with the main distinction being the peak predicted speed on flat ground. None of the existing functions account for the hill slope, which

**Table 2. Final walking speed model variable coefficients.**

|  | *a* | *b* | *c* | *d* |
|---|---|---|---|---|
| Paved road | 1.580 | -0.00389 | -0.00726 | -0.00218 |
| Unpaved road | 1.580 | -0.00389 | -0.00965 | -0.00248 |
| Off-road (obstruction unknown) | 1.536 | -0.00731 | -0.00965 | -0.00187 |
| Off-road (light obstruction) | 1.580 | -0.00731 | -0.00965 | -0.00187 |
| Off-road (heavy obstruction) | 1.443 | -0.00731 | -0.00965 | -0.00187 |

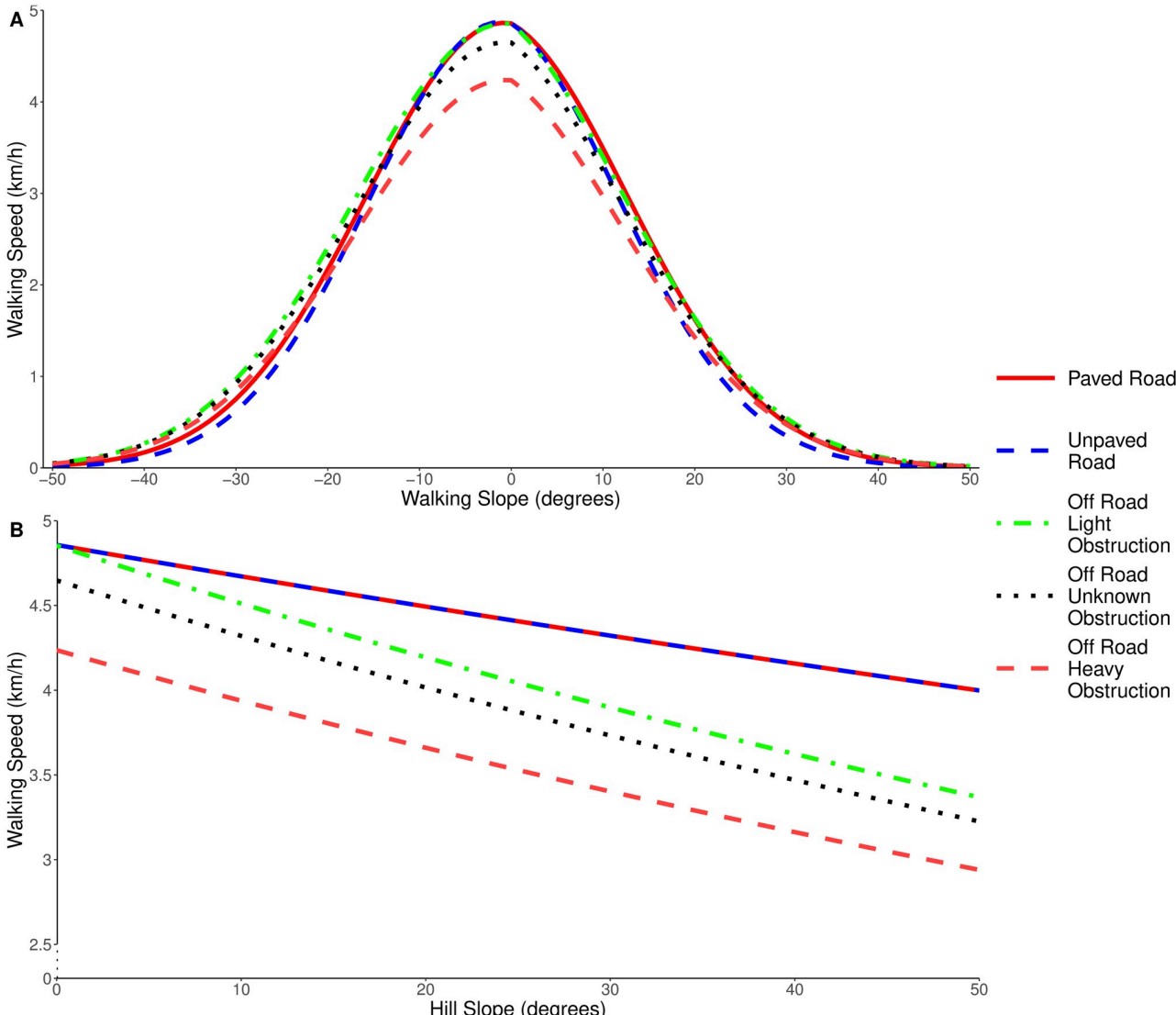

**Fig 3. Walking speed predictions under different terrain conditions.** When: (A) travelling directly up or down hills of varying slope (walking slope), (B) traversing across hills of varying slope (hill slope).

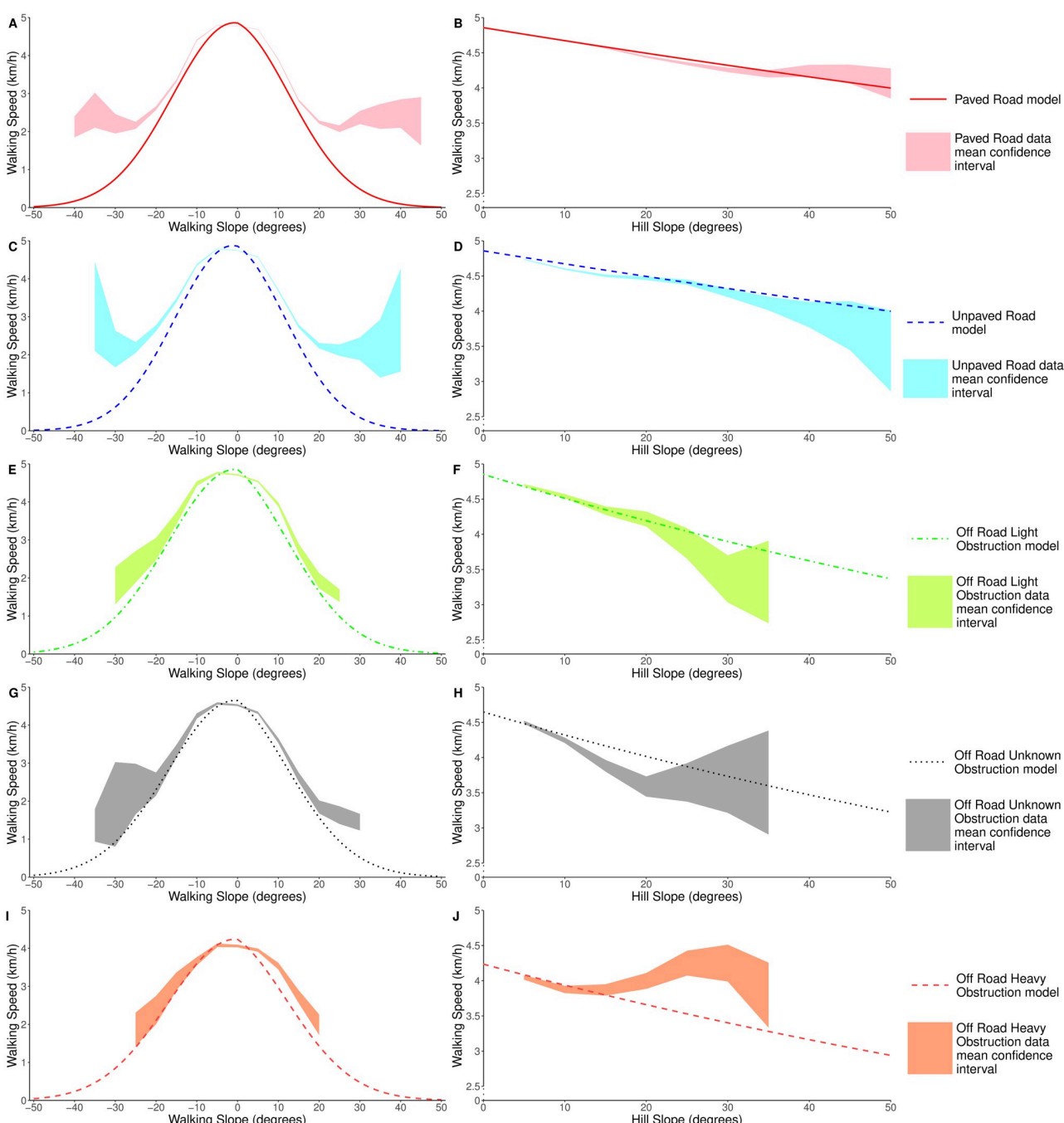

**Fig 4. Walking speed predictions under different terrain conditions.** When: (A,C,E,G,I) travelling directly up or down hills of varying slope (walking slope), (B,D,F,H,J) traversing across hills of varying slope (hill slope). Also shown in each plot is the 95% confidence interval of the mean value of the walking speed for the terrain type, calculated at 5 degree intervals, using data bins with a width of 10 degrees. Note that the confidence intervals were calculated using only data which is within 5 degrees of directly ascending (A,C,E,G,I) or traversing (B,D,F,H,J) the slope.

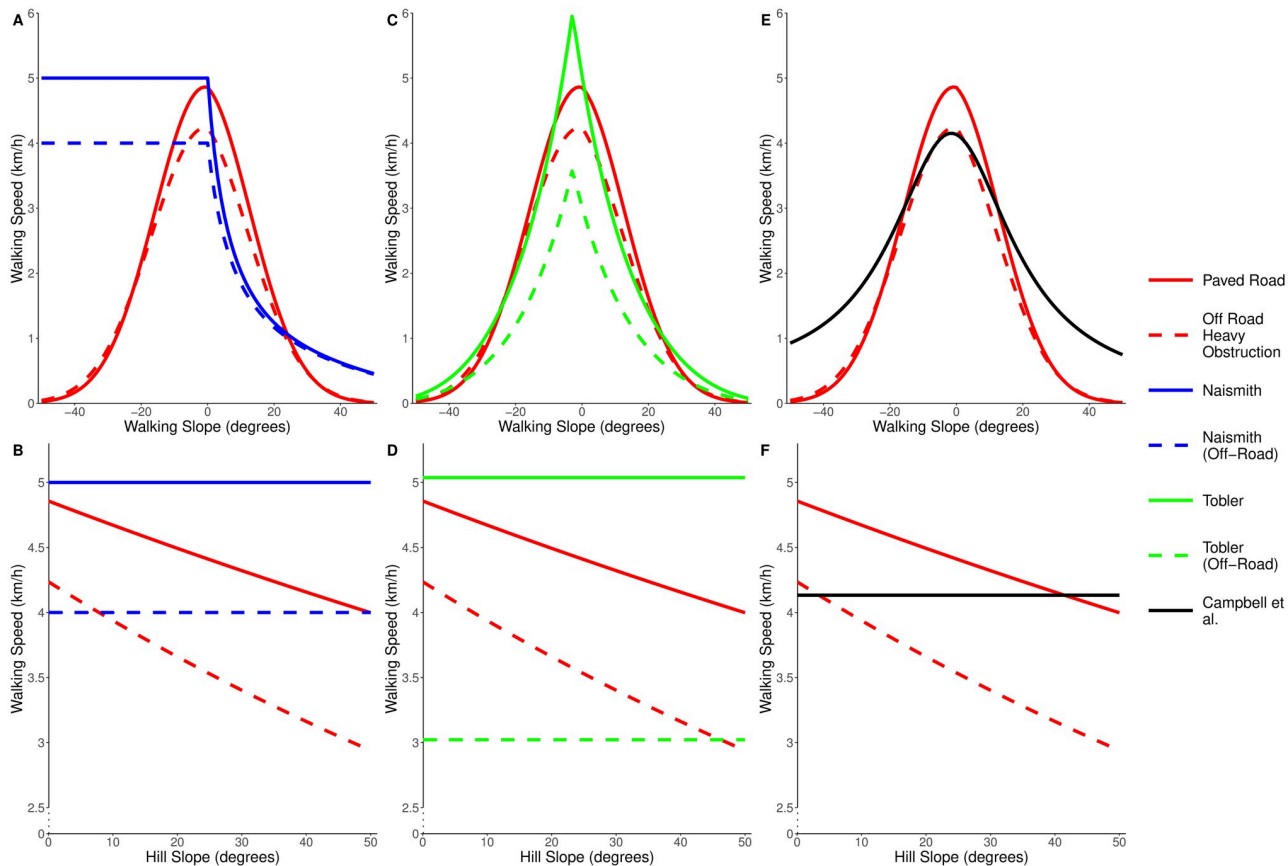

**Fig 5. Comparison of new model and existing hiking functions.** Predicted walking speeds of the new model, Naismith's rule, Tobler's function and Campbell et al.'s function when: (A, C, E) travelling directly up or down hills of varying slope (walking slope), (B, D, F) traversing across hills of varying slope (hill slope).

leads to large disparities when predicting the walking speed for slope traversals. A further example of this can be seen in S6 File, which shows the walking speeds for a simulated off-road route which encounters the full range of hill and walking slopes.

When comparing the performances of each of the models (Table 3), the predicted speeds for individual 50 m sections had a lower RMSE and percentage error, and a higher R squared value using our new model than in the existing ones. The R-squared value is still very low, however we suggest that this is due to the variability within the data. We have previously acknowledged that there are many individual effects which can impact the walking speed, and which we did not attempt to capture in our model. Instead it captures the general trend of the

**Table 3. Comparison of new model against existing methods to calculate walking speeds.**

|  | **New Model** | **Naismith** | **Tobler** | **Campbell** |
|---|---|---|---|---|
| Average % error | 23.68 | 26.36 | 26.17 | 25.33 |
| MSE | 1.20 | 1.61 | 1.53 | 1.58 |
| RMSE | 1.10 | 1.27 | 1.24 | 1.26 |
| $R^2$ | 0.09 | -0.22 | -0.16 | -0.19 |

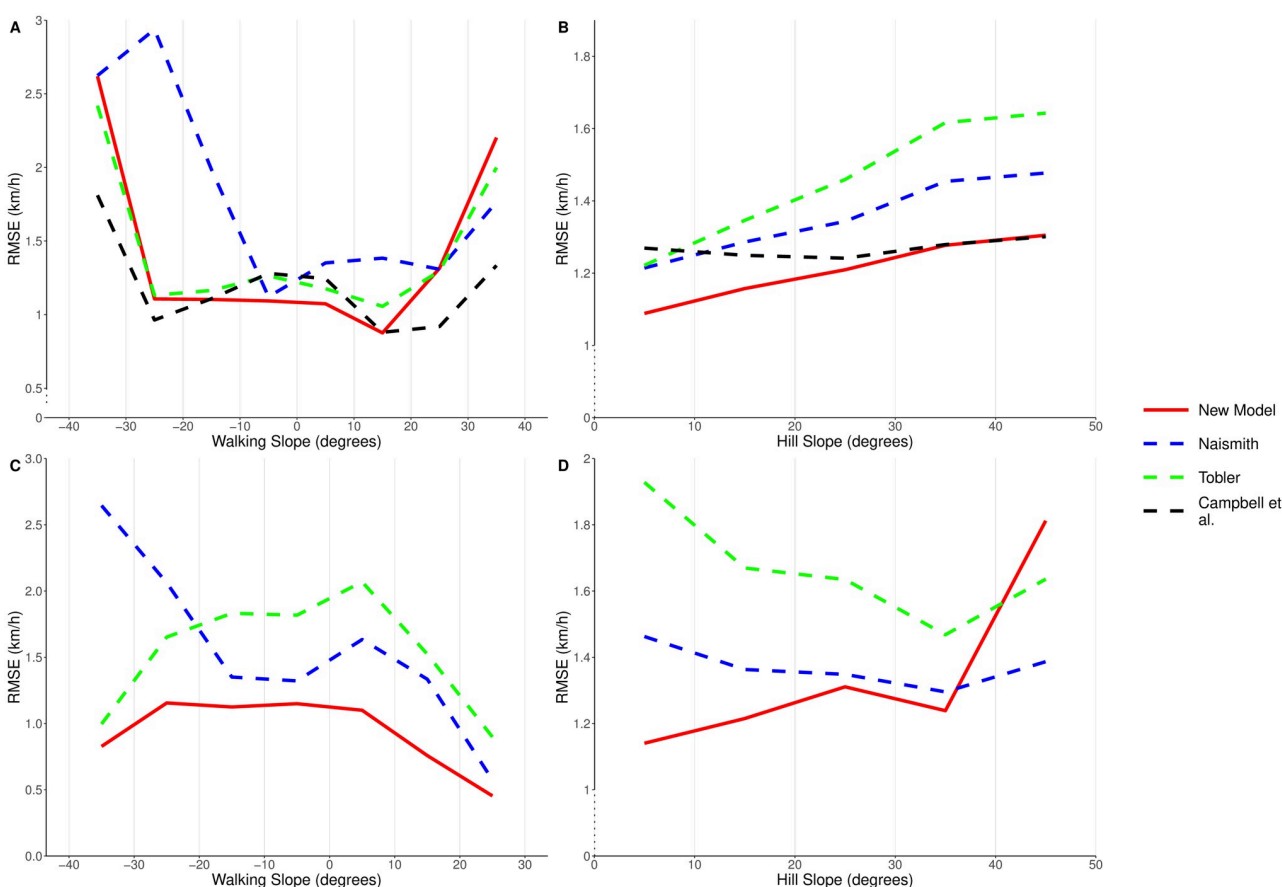

**Fig 6. Comparing RMSE values for the new model, Naismith's rule, Tobler's function and Campbell et al.'s function.** When: (A) travelling directly up or down hills of varying slope (all data, walking slope), (B) traversing across hills of varying slope (all data, hill slope), (C) travelling directly up or down hills of varying slope (off-road data only, walking slope), (D) traversing across hills of varying slope (off-road data only, hill slope). Campbell et al.'s function does not provide off-road speed estimates, so was not included in the off-road data comparisons.

walking speed for an average individual under average conditions, and does this better than existing models (evidenced by the improved RMSE).

To isolate the impact of each of the slope variables, we filtered the results to look at the data where a slope was being directly climbed or traversed. Figs 6A, 6B, 7A and 7B show the RMSE and mean residuals for each of the models, for data which was within 5 degrees of directly climbing (A) or traversing (B) hills of varying slope. From this we can clearly see that Naismith's rule consistently overestimates walking speeds when descending a slope, and underestimates speeds when climbing a slope. When ascending or descending a slope, the RMSE of our GLM is similar to that of Tobler's hiking function. However, one of the main areas where we see an improvement using our model is on slight declines. Tobler's hiking function suggests that walking speed increases on mild descents up to a maximum of 6 km/h. It is clear from Fig 6A, that Tobler's function overestimates the walking speed in this region. Campbell et al.'s function has a slightly lower RMSE value than our new model on the steepest walking slopes, however it underestimates the walking speeds on flat ground and mild slopes; the regions where most walking occurs. Improved walking speed predictions in this region therefore have the greatest impact in real-world situations. Within this region our model consistently has a lower RMSE than the existing functions, and a mean residual error close to 0 km/h.

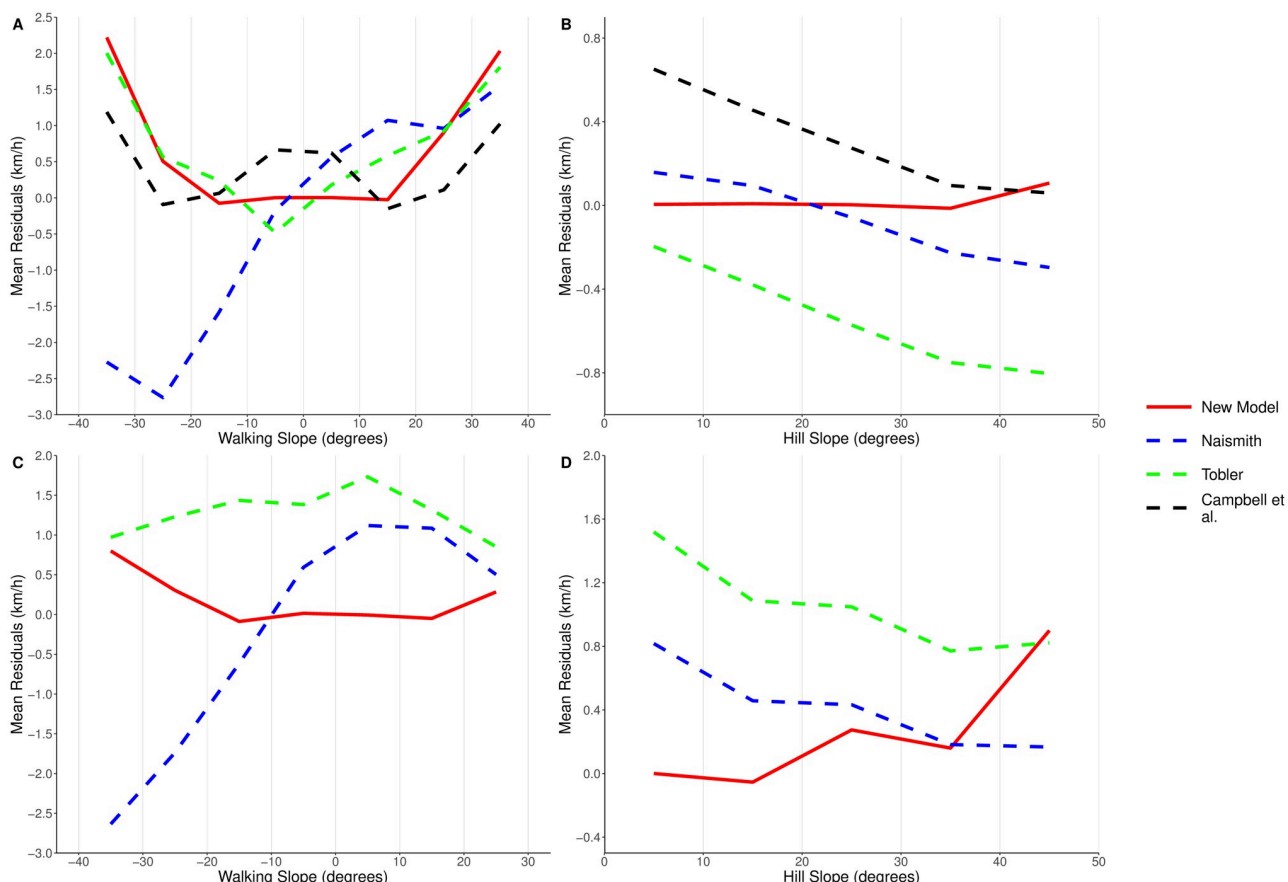

**Fig 7. Comparing mean residual values for the new model, Naismith's rule, Tobler's function and Campbell et al.'s function.** When: (A) travelling directly up or down hills of varying slope (all data, walking slope), (B) traversing across hills of varying slope (all data, hill slope), (C) travelling directly up or down hills of varying slope (off-road data only, walking slope), (D) traversing across hills of varying slope (off-road data only, hill slope). Campbell et al.'s function does not provide off-road speed estimates, so was not included in the off-road data comparisons.

We also see an improvement in RMSE when using our model to predict speeds for hill traversals (Fig 6B). We can note from Fig 7B that both Naismith's rule and Tobler's hiking function consistently overestimate the walking speed when traversing a slope, as they do not take into account the impact that the hill slope has on reducing walking speeds. The performance of Campbell et al's model improves as the hill slope increases, although we suggest this is more due to it underestimating the speed on shallow slopes. We do see that the average error in our model increases as the hill slope increases, but we believe that this is due to limited volumes of data at high hill slopes (∼0.5% of our data occurs on hill slopes steeper than 40 degrees).

As well as looking at the overall performance of our new model, we looked to explore how well our model performed in off-road conditions, compared to the off-road adjustments for the existing functions (Naismith's reduced base speed of 4 km/h, and Tobler's correction factor of 0.6). Figs 6C, 6D, 7C and 7D show the RMSE and mean residuals, only considering data which was recorded in off-road conditions. From Figs 6C and 7C it is clear that Tobler's function consistently underestimates the walking speed when off-road. The factor of 0.6 is a larger reduction in walking speed than is observed in practice. As we found when looking at our data as a whole, Naismith's rule underestimates the walking speed when climbing a slope and

overestimates when descending a slope. Our new model does not suffer from these problems, with both a lower RMSE and lower absolute mean residual value across all walking slopes. Both of these existing models also consistently underestimate walking speeds when traversing a slope, unlike our new model which has a mean residual of less than 0.4 km/h on slopes of up to 35 degrees. The error in predictions of our new model does increase as the hill slope increases, though the RMSE is generally lower than seen in the existing models. On the steepest hill slopes our model appears to perform less well than the existing ones, though only 0.2% of our off-road data occurred on a hill slope steeper than 40 degrees.

Although we have shown an improvement in walking speed predictions over short sections of routes, this did not translate to similar results when looking at predicted walking times for routes as a whole. Our model and all of the existing models which we have explored here had an average percentage error of 13.5%—15.5% when predicting the time taken for a complete route. However, based on the errors seen in Figs 6 and 7, we believe that this is a result of errors cancelling out over the course of a hike. For example while ascending a hill, Naismith's rule will underestimate the walking speed (and thus overestimate the walking time), but it will then overestimate the walking speed on the subsequent descent, leading to a relatively accurate total time estimate. The results here suggest that Naismith's rule, and other existing functions, are still a good rule of thumb to calculate route times as a whole, but time estimates for individual sections of a route will be less accurate than when using the new model found here.

## Discussion

We have developed a model for walking speed which is very robust, due the large volume of data (88,000 km) used to build it, and which correlates with the data over a wider range of conditions than commonly used formulae. Data from tracks confirms that each of the walking slope, the hill slope and the terrain type or obstruction are significant factors in determining walking speeds. The model improves on existing methods to predict walking speeds (Figs 6 & 7). We have also shown the specific improvement that our new model has on predicting walking speeds in off-road conditions, compared to the simple off-road speed reductions used by existing models. The existing methods to calculate walking speeds require tuning for use in real-world scenarios, as there are many factors which can affect an individual's walking speed beyond the slope and obstruction level (such as the weather, fitness level or age) [2–4]. The model presented here requires the same tuning as these existing methods but provides more a more accurate population average walking speed across a wide range of terrain and slope conditions.

Our results confirm that Naismith's rule (Fig 1) is still a good rule-of-thumb to use when estimating the total walking time for a route, especially in situations where the calculation must be done by hand. However, the findings here can be used as an addition to Naismith's rule; it is likely that (under Naismith's rule) the predicted ascent time will be overestimated and the predicted descent time will be underestimated. It is not uncommon for hikers to contact one another when they reach the summit of a hill, and provide an estimated arrival time back at the campsite. Knowing that the descent will likely take longer than estimated by Naismith's rule will result in more accurate arrival estimations being given. Similarly, the knowledge of how the hill slope reduces walking speeds, or that just 10 cm of vegetation can reduce walking speeds by up to 0.6 km/h may well affect route choices made when out on a walk. For example, if a hiker is following a footpath, but can see from their map that the path forms a large curve then they can use our findings to decide whether it will be faster to travel off-road and cut the corner. On flat terrain with heavy levels of obstruction, our model suggests that such a short cut will be faster if the distance covered on the path is more than 15% longer than

the off-road distance. Speed is not the only factor which would affect this decision, as safety and navigability are also important variables, but these results can help people make more informed choices when on a hike.

The benefit of using crowdsourced GPS data to build our model is also a limitation of the approach, as we did not have control over data collection. This meant that models were unable to account for any bias in our data such as group size, ability and composition, or other potential variables such as weather conditions, as factors in determining walking speed (although we would expect the volume of data to cause most of these effects to average out).

Unlike previous work [8], we did not use fixed values to classify breaks and non-walking or hiking tracks. Instead we developed filters based on the attributes of known walking data (see S2 File). The methods used to filter the datasets were blinded to the outcome of the model generation, the choice of filtering methods will have had an impact on the dataset and subsequent model and no ground truth was available against which to test our assumptions.

Our method of calculating the terrain obstruction value was relatively crude, looking only at the obstruction height at each GPS point. While this did prove to be successful, and we observed a clear difference in walking speeds between areas of light and heavy obstruction (see S5 File), the inaccuracies present within GPS data may have led to some erroneous obstruction measurements, for example in a field sparsely populated with trees. In future, efforts should be made to refine this approach, such as considering the average obstruction level over a wider area around each point.

A further limitation of our data came when we looked to classify points into paved roads, unpaved roads or off-road. A combination of GPS drift and map error means that there is significant uncertainty and so we had to use a search radius around each data point to identify potential roads. We suspect that we were likely overclassifying tracks on roads. While our model appears to be robust to this overclassification (due to the volumes of correctly classified data used), the overclassification left us with a reduced number of off-road datapoints from which to predict off-road travel speeds.

Furthermore, the use of crowdsourced data meant that all of our data came from 'walkable' regions by definition. When including the terrain obstruction variable, we were unable to determine if there are levels of terrain obstruction which makes walking impossible. Similarly, the vast majority of the data was collected on shallow hill- and walking slopes, leading to a sparcity of data in steeper areas. While this does mean that we can be very confident about our walking speed predictions in less steep regions (where most walking occurs), it is unclear whether the lack of data on steeper regions is a result of steep slopes being relatively rare, or that they cannot be easily navigated, so hikers chose an alternate path. As described above we had to make a number of assumptions regarding data filtering and processing including model selection, and other choices may give different results. To support anyone who wants to challenge or test these assumptions, or try different models, we have made all our code available on Github. Further, all of the data sources used are detailed in S1 File and the filters/ assumptions we used to clean the data are fully detailed in S2 File.

## Conclusion

Widely used algorithms (e.g. Naismith's rule) for estimating walking/hiking speed are simple to understand, very easy to calculate but are based on limited observations. Here we curated a dataset of almost 88,000 km of walking and hiking data. We found that the existing algorithms perform quite well against the dataset but they tend to overestimate ascent time, underestimate descent time and most ignore terrain obstruction and hill slope both of which we found to be significant factors. We used the data to derive a new model that takes into account these

variables. We demonstrated that the model provides more accurate walking speeds than the existing methods in all scenarios, and particularly in off-road regions. By providing improved walking speed predictions in these off-road regions, we have enabled more accurate calculations of the fastest route to or from any given location, which could save minutes in an emergency situation where every second is important.

## Supporting information

**S1 File. Data sources.**
(PDF)

**S2 File. Data acquisition and preparation.**
(PDF)

**S3 File. Exploratory data modelling study.**
(PDF)

**S4 File. Exploring the differences between Scotland and the rest of the UK.**
(PDF)

**S5 File. Exploring the impact of terrain obstruction.**
(PDF)

**S6 File. Comparison of walking speed changes while crossing a simulated off-road terrain region.**
(PDF)

## Acknowledgments

Preprocessing of the GPX files made use of the resources provided by the Edinburgh Compute and Data Facility (ECDF) [34].

## Author Contributions

**Conceptualization:** Andrew Wood.

**Data curation:** Andrew Wood.

**Formal analysis:** Andrew Wood.

**Investigation:** Andrew Wood.

**Methodology:** Andrew Wood, T. Ian Simpson, J. Douglas Armstrong.

**Resources:** Andrew Wood.

**Supervision:** William Mackaness, J. Douglas Armstrong.

**Writing – original draft:** Andrew Wood.

**Writing – review & editing:** Andrew Wood, William Mackaness, T. Ian Simpson, J. Douglas Armstrong.

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
