## [Decision Letter · Decision Letter 0]

24 Jul 2023

PONE-D-23-19311Improved prediction of hiking speeds using a data driven approachPLOS ONE

Dear Dr. Wood,

Thank you for submitting your manuscript to PLOS ONE. After careful consideration, we feel that it has merit but does not fully meet PLOS ONE’s publication criteria as it currently stands. Therefore, we invite you to submit a revised version of the manuscript that addresses the points raised during the review process.

We look forward to receiving your revised manuscript.

Kind regards,

Yuxia Wang

Academic Editor

PLOS ONE

Journal Requirements:

   "This work was funded by the UK Engineering and Physical Sciences Research Council (grant EP/R513209/1) and the University of Edinburgh. It was supported in data acquisition by Ordnance Survey and Digimap.Preprocessing of the GPX files made use of the resources provided by the Edinburgh Compute and Data Facility (ECDF)"

   "Author Andrew Wood funded by the UK Engineering and Physical Sciences Research Council (grant EP/R513209/1), https://www.ukri.org/councils/epsrc/

4. We note that Figure 2 in your submission contain map/satellite images which may be copyrighted. All PLOS content is published under the Creative Commons Attribution License (CC BY 4.0), which means that the manuscript, images, and Supporting Information files will be freely available online, and any third party is permitted to access, download, copy, distribute, and use these materials in any way, even commercially, with proper attribution. For these reasons, we cannot publish previously copyrighted maps or satellite images created using proprietary data, such as Google software (Google Maps, Street View, and Earth). For more information, see our copyright guidelines: http://journals.plos.org/plosone/s/licenses-and-copyright.

Reviewers' comments:

Reviewer's Responses to Questions

**Comments to the Author**

1. Is the manuscript technically sound, and do the data support the conclusions?

Reviewer #1: Partly

Reviewer #2: Yes

2. Has the statistical analysis been performed appropriately and rigorously? 

Reviewer #1: Yes

Reviewer #2: Yes

3. Have the authors made all data underlying the findings in their manuscript fully available?

Reviewer #1: Yes

Reviewer #2: Yes

4. Is the manuscript presented in an intelligible fashion and written in standard English?

Reviewer #1: Yes

Reviewer #2: Yes

5. Review Comments to the Author

Reviewer #1: This research centers on the prediction of hiking speed using a novel generalized linear model. By integrating public GPS data, this model has the ability to forecast walking speed by considering the gradient of the terrain (hill slope) and the degree of terrain obstruction. Despite the announcement of substantial improvement compared to various established models, there are several revisions that need to be addressed first.

1. The methods of the paper need further clarification. First, detailed descriptions of the data and data processing should be included in the main body of the paper, rather than in the Supporting Information, as this is a paper focusing on a data-driven approach. This leads to the structure of the paper being fragmented and some terminologies sound wired and confusing. For instance, what is meant by "datapoint" mentioned in line 152?

2. Additionally, more details regarding the Generalised Linear Model, which forms the foundation of the research, should be provided. Questions that need to be addressed include: What are the inputs? How are these variables organized? What is the relationship between variables a, b, c, d and the results presented in Table 1? How does the critical gradient affect the performance of the model? Are the speed and characteristics aggregated at a point level or in line string segments?

3. The introduction section of the document is excessively lengthy and would benefit from a reorganization of its content. I suggest incorporating some of the introductory material about the hiking speed model into the methods section. Furthermore, the introductory material concerning the hiking speed model should be summarized (in introduction section) and present the formulations instead of listing various models and qualitatively describe these models (in method section). This will facilitate the comprehension of the paper.

4. Considering that the model utilizes multiple variables and employs GLM for regression, the improvement achieved by the model is not as significant, which could potentially impact its practical application. While the model is expected to outperform rule-based models that do not fit the empirical data in previous studies, I have reservations about its applicability in real-life scenarios. The author should provide more details about the results and discuss the implications of these performance changes on hiking activities.

5. In terms of the conclusion, the research should interpret the social implications of the findings derived from the data. The importance of hiking speed estimations and the contributions of the paper should be emphasized in both introduction and conclusion.

Reviewer #2: Summary

In this paper, Wood et al. propose a new set of equations for predicting travel rates as a function of landscape conditions. Whereas several functions exist for the prediction of rates driven by walking slope (slope in the direction of pedestrian travel), few take into consideration the hill slope (slope in the direction of the terrain’s steepest descent) and/or the presence of terrain obstructions above the ground surface. Wood et al. incorporate both of these two characteristics, finding significant effects of each and providing novel quantitative insight that could be valuable for more robust travel rate predictions, particularly in off-road/off-trail environments. The paper is well presented, and the topic is of wide interdisciplinary interest – well-suited for PLOS ONE. I have a few major and minor concerns that I believe should be addressed prior to publication, but I do believe the work will eventually be a valuable contribution to the existing literature.

Major Comments

- It seems terrain obstructions could be handled in a more elegant way. Simply using height in a binary fashion as the basis of determining off-road impedance seems like a missed opportunity. Height is definitely one important consideration (it’s easier to step over short vegetation than tall vegetation), but density is another arguably more important one. One can easily walk through tall but sparse vegetation just as one can easily walk through short but dense vegetation. Some focal measure of density (e.g., the number of lidar-derived pixels above a certain height threshold within a given neighborhood) would be worth examining.

- I would like to see comparisons to more existing travel rate functions. Several papers have already demonstrated superiority over Tobler and Naismith. To ensure that this work truly represents a valuable contribution to the literature requires comparisons to more contemporary algorithms. Although it’s not necessarily the most important statistical measure of model performance in the context of GLMs, an R-squared value of 0.09 does not leave the reader with a high degree of confidence in your new model.

- Related to the point above, you point out the travel rates are largely just a means to getting at a more useful measure in travel time. You provide several examples about the importance of estimating travel time. Can you demonstrate how your new algorithm provides an opportunity for the accurate estimation of travel time? I think the results would look a lot better than your travel rate estimates, since predicting time over a longer hike should have smaller margins of error than instantaneous travel rates derived from erroneous GNSS data.

- I think a lot of valuable information that could be included in the main manuscript is placed in the supplementary materials. For example, the main manuscript does not have any depiction of the final function forms (line plots of speed vs. slope, e.g., S4 Fig 3), which seems like an important omission. Also S3 Fig 1 provides really useful insight into the complexity of trying to predict travel rates, given the extreme variability in the data.

Minor Comments

L4: Extremely minor point but I’m not sure I would consider it “standard practice”, per se. It’s certainly *good* practice.

L2-8: You begin the paragraph stating that travel rates are important in “many situations” and then only proceed to give one example. You might consider adding more to frame the importance of your study.

L14-15: I appreciate the split into individual and external factors, but to say that the effects of slope, for example, “will be consistent across all individuals” isn’t true. Slope, and other landscape conditions, affects people very differently.

L2-23: The first three paragraphs have not a single reference to existing literature. I’ll grant you that a lot of the discussion up to this point is based on intuition/experience, but there are several statements that would carry more weight with references to existing studies.

L33-34: I don’t understand this statement. Are you suggesting that Naismith’s rule does not account for slope? That’s *precisely* what Naismith’s rule does… Please clarify/rephrase.

L67: very different situations than what?

L84 and throughout: You might consider the more universal term of global navigation satellite systems (GNSS) rather than the US-specific Global Positioning System (GPS)

L94 and elsewhere: It seems you should define “terrain obstruction” explicitly. I assume this means the presence of vegetation, primarily? But I could also imagine a cliff being considered a terrain obstruction.

Figure 2 caption: should read “can *be* identified”

L145-151: So speeds are based solely on time between sequential GNSS positions and the timing of those same positions? Wouldn’t this provide an underestimate of speeds? I’m picturing a windy, zig-zagging trail… If a GNSS position is only recorded every, say, 30 seconds, then the resulting “track” may appear to move straight through the zigzags, giving an underestimate of the distance actually traveled.

L174: more important than the version of RStudio (simply the IDE) is the version of R

S2

In the Break Finding section, about 2/3 through the second paragraph, you say “…paused recording for a break – example here”. Is this an error?

6. PLOS authors have the option to publish the peer review history of their article (what does this mean?). If published, this will include your full peer review and any attached files.

Reviewer #1: No

Reviewer #2: No

---

## [Author Response · Author response to Decision Letter 0]

7 Sep 2023

• The paper was prepared using the Plos Latex Template (https://www.overleaf.com/latex/templates/latex-template-for-plos-public-library-of-science-articles/wdmgcwzgvhnn) on Overleaf, so we believe it meets the requirements. Please let us know if further changes are required.

 "This work was funded by the UK Engineering and Physical Sciences Research Council (grant EP/R513209/1) and the University of Edinburgh. It was supported in data acquisition by Ordnance Survey and Digimap.Preprocessing of the GPX files made use of the resources provided by the Edinburgh Compute and Data Facility (ECDF)"

 "Author Andrew Wood funded by the UK Engineering and Physical Sciences Research Council (grant EP/R513209/1), https://www.ukri.org/councils/epsrc/

• We have removed the funding source from the acknowledgements section. The current funding statement is correct.

• It is not possible to share the data directly, as it is under copyright. Data sources and a detailed methodology to extract the data from original sources to reproduce the work have been provided in the Supplementary Information. This has been confirmed by the reviewers in their response to Question 3.

4. We note that Figure 2 in your submission contain map/satellite images which may be copyrighted. All PLOS content is published under the Creative Commons Attribution License (CC BY 4.0), which means that the manuscript, images, and Supporting Information files will be freely available online, and any third party is permitted to access, download, copy, distribute, and use these materials in any way, even commercially, with proper attribution. For these reasons, we cannot publish previously copyrighted maps or satellite images created using proprietary data, such as Google software (Google Maps, Street View, and Earth). For more information, see our copyright guidelines: http://journals.plos.org/plosone/s/licenses-and-copyright.

• Figure 2 has been changed to use imagery from OpenStreetMap which is available under the Open Data Commons Open Database License

5. Review Comments to the Author

Reviewer #1: This research centers on the prediction of hiking speed using a novel generalized linear model. By integrating public GPS data, this model has the ability to forecast walking speed by considering the gradient of the terrain (hill slope) and the degree of terrain obstruction. Despite the announcement of substantial improvement compared to various established models, there are several revisions that need to be addressed first.

1. The methods of the paper need further clarification. First, detailed descriptions of the data and data processing should be included in the main body of the paper, rather than in the Supporting Information, as this is a paper focusing on a data-driven approach. This leads to the structure of the paper being fragmented and some terminologies sound wired and confusing. For instance, what is meant by "datapoint" mentioned in line 152?

• Algorithms explaining the data processing steps have been added to the main body of the paper (pages 6-7). Additional minor changes have been added to the Methods to provide more explanation

2. Additionally, more details regarding the Generalised Linear Model, which forms the foundation of the research, should be provided. Questions that need to be addressed include: What are the inputs? How are these variables organized? What is the relationship between variables a, b, c, d and the results presented in Table 1? How does the critical gradient affect the performance of the model? Are the speed and characteristics aggregated at a point level or in line string segments?

• Table 1 has changed to better reflect the GLM. The critical gradient does not affect the performance of the model, other than that the model was selected from those which put the critical gradient in the correct region. This has been added to the main body of the paper (Lines 179-183). 

3. The introduction section of the document is excessively lengthy and would benefit from a reorganization of its content. I suggest incorporating some of the introductory material about the hiking speed model into the methods section. Furthermore, the introductory material concerning the hiking speed model should be summarized (in introduction section) and present the formulations instead of listing various models and qualitatively describe these models (in method section). This will facilitate the comprehension of the paper.

• The introduction has been reduced in length, removing extraneous information. Information regarding the critical gradient and data has been moved to the materials and methods section 

4. Considering that the model utilizes multiple variables and employs GLM for regression, the improvement achieved by the model is not as significant, which could potentially impact its practical application. While the model is expected to outperform rule-based models that do not fit the empirical data in previous studies, I have reservations about its applicability in real-life scenarios. The author should provide more details about the results and discuss the implications of these performance changes on hiking activities.

• Due to the crowdsourced nature of our dataset, we do not have a single route which has been recorded by multiple individuals. This makes time comparison to a real route difficult, due to the variance within walking speeds. We could cherrypick a route for which our model outperforms the existing ones, but have no way of knowing if that user was walking at the true population average speed, so feel that this would not further readers’ understanding.

• We have added a discussion on travel time over routes as a whole (where all models perform equally), and suggested that this is due to errors cancelling out (Lines 310-321, 333-339) We believe that the main strength of the new model is shown when predicting the walking speed and time for individual sections of a route. 

• A simulation of a hike section has been added as S6 Supporting Information, which shows how the different models predict walking speeds while crossing a simulated hill. Walking time estimates for this simulated route are not given, as there is no way to know which of the models comes closest to predicting the true walking time.

• Hypothetical scenarios of situations where the new model provides an improvement over the current knowledge, and can impact decisions made while walking have been added to the discussion (Lines 341-348).

5. In terms of the conclusion, the research should interpret the social implications of the findings derived from the data. The importance of hiking speed estimations and the contributions of the paper should be emphasized in both introduction and conclusion.

• More examples of when the hiking time is important have been added to the introduction (Lines 8-13) The discussion has also been added with hypothetical scenarios of situations where the new model provides an improvement over the current knowledge (Lines 341-348).

Reviewer #2: Summary

In this paper, Wood et al. propose a new set of equations for predicting travel rates as a function of landscape conditions. Whereas several functions exist for the prediction of rates driven by walking slope (slope in the direction of pedestrian travel), few take into consideration the hill slope (slope in the direction of the terrain’s steepest descent) and/or the presence of terrain obstructions above the ground surface. Wood et al. incorporate both of these two characteristics, finding significant effects of each and providing novel quantitative insight that could be valuable for more robust travel rate predictions, particularly in off-road/off-trail environments. The paper is well presented, and the topic is of wide interdisciplinary interest – well-suited for PLOS ONE. I have a few major and minor concerns that I believe should be addressed prior to publication, but I do believe the work will eventually be a valuable contribution to the existing literature.

Major Comments

- It seems terrain obstructions could be handled in a more elegant way. Simply using height in a binary fashion as the basis of determining off-road impedance seems like a missed opportunity. Height is definitely one important consideration (it’s easier to step over short vegetation than tall vegetation), but density is another arguably more important one. One can easily walk through tall but sparse vegetation just as one can easily walk through short but dense vegetation. Some focal measure of density (e.g., the number of lidar-derived pixels above a certain height threshold within a given neighborhood) would be worth examining.

• Terrain obstruction had not been explored for its impact on walking speeds prior to this work. During exploration we found that the simple metric of obstruction height is very significant at predicting the walking speed. We acknowledge that this is a potential limitation and avenue for further research and have added this to the discussion (Lines 361-367).

- I would like to see comparisons to more existing travel rate functions. Several papers have already demonstrated superiority over Tobler and Naismith. To ensure that this work truly represents a valuable contribution to the literature requires comparisons to more contemporary algorithms. Although it’s not necessarily the most important statistical measure of model performance in the context of GLMs, an R-squared value of 0.09 does not leave the reader with a high degree of confidence in your new model.

• Other methods have not gained widespread use, likely due to the very small sample sizes which they were based on. A comparison with the most up to date work by Campbell et al, which also uses crowdsourced data, has been added throughout (Figs 1,4,5,6 and corresponding text)

- Related to the point above, you point out the travel rates are largely just a means to getting at a more useful measure in travel time. You provide several examples about the importance of estimating travel time. Can you demonstrate how your new algorithm provides an opportunity for the accurate estimation of travel time? I think the results would look a lot better than your travel rate estimates, since predicting time over a longer hike should have smaller margins of error than instantaneous travel rates derived from erroneous GNSS data.

• Due to the crowdsourced nature of our dataset, we do not have a single route which has been recorded by multiple individuals. This makes time comparison to a real route difficult, due to the variance within walking speeds. We could cherrypick a route for which our model outperforms the existing ones, but have no way of knowing if that user was walking at the true population average speed, so feel that this would not further readers’ understanding.

• We have added a discussion on travel time over routes as a whole (where all models perform equally), and suggested that this is due to errors cancelling out (Lines 310-321, 333-339) We believe that the main strength of the new model is shown when predicting the walking speed and time for individual sections of a route. 

• A simulation of a hike section has been added as S6 Supporting Information, which shows how the different models predict walking speeds while crossing a simulated hill. Walking time estimates for this simulated route are not given, as there is no way to know which of the models comes closest to predicting the true walking time.

• Hypothetical scenarios of situations where the new model provides an improvement over the current knowledge, and can impact decisions made while walking have been added to the discussion (Lines 341-348).

- I think a lot of valuable information that could be included in the main manuscript is placed in the supplementary materials. For example, the main manuscript does not have any depiction of the final function forms (line plots of speed vs. slope, e.g., S4 Fig 3), which seems like an important omission. Also S3 Fig 1 provides really useful insight into the complexity of trying to predict travel rates, given the extreme variability in the data.

• Depictions of the final function forms have been added to the main body of the paper (Fig 3), as have Algorithms detailing the breakfinding and data filtering processes (pages 6,7)

Minor Comments

L4: Extremely minor point but I’m not sure I would consider it “standard practice”, per se. It’s certainly *good* practice.

• Changed to good practice and added citation (Line 4)

L2-8: You begin the paragraph stating that travel rates are important in “many situations” and then only proceed to give one example. You might consider adding more to frame the importance of your study.

• Further examples have been added (Lines 8-13)

L14-15: I appreciate the split into individual and external factors, but to say that the effects of slope, for example, “will be consistent across all individuals” isn’t true. Slope, and other landscape conditions, affects people very differently.

• We were trying to suggest that the variable will be consistent (ie the slope will always be that steep), rather than the effect of the variable. This has been changed to try and make this clearer (Lines 19-20)

L2-23: The first three paragraphs have not a single reference to existing literature. I’ll grant you that a lot of the discussion up to this point is based on intuition/experience, but there are several statements that would carry more weight with references to existing studies.

• References have been added to back up some of the statements (citations 1 & 2, lines 6,16)

L33-34: I don’t understand this statement. Are you suggesting that Naismith’s rule does not account for slope? That’s *precisely* what Naismith’s rule does… Please clarify/rephrase.

• Naismiths rule accounts for slope when walking uphill, but does not when walking downhill. This has been rephrased to make it clearer. (Lines 38-40)

L67: very different situations than what?

• This has been changed to emphasise that the participants were running and not walking (Line 60)

L84 and throughout: You might consider the more universal term of global navigation satellite systems (GNSS) rather than the US-specific Global Positioning System (GPS)

• GNSS has been added (line 77), with a note that GPS is the more frequently used term by the general public

• We chose to use GPS throughout the rest of the body of the paper, as it is the more widespread term, so felt that the paper is more accessible to a wider audience by using it

L94 and elsewhere: It seems you should define “terrain obstruction” explicitly. I assume this means the presence of vegetation, primarily? But I could also imagine a cliff being considered a terrain obstruction.

• A definition of terrain obstruction has been included (Line 116)

Figure 2 caption: should read “can *be* identified”

• Thank you, this has been changed

L145-151: So speeds are based solely on time between sequential GNSS positions and the timing of those same positions? Wouldn’t this provide an underestimate of speeds? I’m picturing a windy, zig-zagging trail… If a GNSS position is only recorded every, say, 30 seconds, then the resulting “track” may appear to move straight through the zigzags, giving an underestimate of the distance actually traveled.

• In practice, the data positions are generally recorded every 3-5 seconds as the device detects movement, so this is not an issue

L174: more important than the version of RStudio (simply the IDE) is the version of R

• Thank you, this has been changed (Line 188)

S2

In the Break Finding section, about 2/3 through the second paragraph, you say “…paused recording for a break – example here”. Is this an error?

• Thank you, this has been changed

---

## [Decision Letter · Decision Letter 1]

9 Oct 2023

PONE-D-23-19311R1Improved prediction of hiking speeds using a data driven approachPLOS ONE

Dear Dr. Wood,

Thank you for submitting your manuscript to PLOS ONE. After careful consideration, we feel that it has merit but does not fully meet PLOS ONE’s publication criteria as it currently stands. Therefore, we invite you to submit a revised version of the manuscript that addresses the points raised during the review process.

We look forward to receiving your revised manuscript.

Kind regards,

Yuxia Wang

Academic Editor

PLOS ONE

Additional Editor Comments:

Dear Authors,

We received two reviews of your manuscript. While Reviewer #2 recommended acceptance, Reviewer #1 pointed some significant concerns which need to be addressed. Therefore, I would like to suggestion a major revision. During the revision, please pay attention to the suggestion and comments of Reviewer #1. Please note that your revised version will be further assessed by external reviewers.

Reviewers' comments:

Reviewer's Responses to Questions

**Comments to the Author**

1. If the authors have adequately addressed your comments raised in a previous round of review and you feel that this manuscript is now acceptable for publication, you may indicate that here to bypass the “Comments to the Author” section, enter your conflict of interest statement in the “Confidential to Editor” section, and submit your "Accept" recommendation.

Reviewer #1: (No Response)

Reviewer #2: All comments have been addressed

2. Is the manuscript technically sound, and do the data support the conclusions?

Reviewer #1: Yes

Reviewer #2: Yes

3. Has the statistical analysis been performed appropriately and rigorously? 

Reviewer #1: Yes

Reviewer #2: Yes

4. Have the authors made all data underlying the findings in their manuscript fully available?

Reviewer #1: Yes

Reviewer #2: (No Response)

5. Is the manuscript presented in an intelligible fashion and written in standard English?

Reviewer #1: Yes

Reviewer #2: Yes

6. Review Comments to the Author

Reviewer #1: The revised version of the paper has undergone substantial improvements in terms of its writing, logic, and clarity. However, there remain some significant concerns that, in my opinion, should be addressed before publication.

Statistical Significance of Model Improvement: One critical concern, previously highlighted in the last round of review comments, pertains to the practical significance of the model's performance. Given that the model incorporates multiple variables and employs Generalized Linear Models (GLM) for regression, the reported improvement achieved by the model appears to be modest. This issue has also been noted by other reviewers, particularly concerning the low R-squared value of 0.09. The authors should not overlook this concern and should provide a more thorough discussion of the model's practical utility and limitations.

Detailed Presentation of Data-Driven Approach: As the paper focuses on a data-driven approach to modeling walking speed, it would greatly enhance the understanding of the methodology if the authors provided more quantitative characteristics of the original data. For instance, Figure 3 illustrates walking speed predictions under various terrain conditions. It is essential to include statistical characteristics of the actual speed data (e.g., average, standard deviation, or the 95% confidence interval) when individuals are traversing different types of terrain, derived from real datasets. Additionally, incorporating statistical characteristics of the ground truth data in Figure 4 would strengthen the paper's credibility and the evaluation of the model's performance.

Transparency in Exclusion of Data Points: It would be beneficial for readers to know the number and percentage of data points or sections that were excluded due to the filtering process. This transparency will provide a clearer picture of the data selection and processing steps.

Additional Minor Comments:

Introduction References: In the introduction section, consider providing more references when discussing the classification of factors that can impact walking speed into two groups. This will enhance the depth of the literature review and provide a stronger foundation for your work.

Model Performance Evaluation: Highlighting the fact that the model's performance evaluations are conducted at the section level should be done earlier in the paper to prevent any misunderstanding. Additionally, consider presenting statistical characteristics of the sections, such as their length and the distribution of slopes, to provide a comprehensive assessment.

Clarify Terminology: Distinguish between "hill slope" and "walking slope" throughout the paper to avoid confusion. For instance, in the caption for Figure 3, specify the meaning of "slope" in Figures 3A and 3B to enhance clarity for readers.

Overall, while the revisions have improved the paper's quality, addressing these concerns and making the suggested enhancements will further enhance the paper's scientific rigor and comprehensibility.

Reviewer #2: I thank the authors for addressing my comments thoroughly and am happy to recommend that this revised version be accepted for publication. It will be a valuable contribution to the literature.

7. PLOS authors have the option to publish the peer review history of their article (what does this mean?). If published, this will include your full peer review and any attached files.

Reviewer #1: No

Reviewer #2: No

---

## [Author Response · Author response to Decision Letter 1]

22 Nov 2023

We would like to thank the reviewers for their additional comments and appreciate the improvements that these have made to the paper.

Statistical Significance of Model Improvement: One critical concern, previously highlighted in the last round of review comments, pertains to the practical significance of the model's performance. Given that the model incorporates multiple variables and employs Generalized Linear Models (GLM) for regression, the reported improvement achieved by the model appears to be modest. This issue has also been noted by other reviewers, particularly concerning the low R-squared value of 0.09. The authors should not overlook this concern and should provide a more thorough discussion of the model's practical utility and limitations.

The low R-squared indicates that there remains variability in the data that is not captured in the model. We have acknowledged this from the beginning by indicating that we know of many variables which will impact the speed (such as group ability etc), for which there are no data available to include in the model. This has been made more explicit at the end of the introduction [lines 99-101]. Further acknowledgement of this has also been added to the results [lines 288-292], and the following has been added to the discussion [Lines 356-361]: “The existing methods to calculate walking speeds require tuning for use in real-world scenarios, as there are many factors which can affect an individual's walking speed beyond the slope and obstruction level (such as the weather, fitness level or age). The model presented here requires the same tuning as these existing methods but provides more a more accurate population average walking speed across a wide range of terrain and slope conditions.”

Detailed Presentation of Data-Driven Approach: As the paper focuses on a data-driven approach to modeling walking speed, it would greatly enhance the understanding of the methodology if the authors provided more quantitative characteristics of the original data. For instance, Figure 3 illustrates walking speed predictions under various terrain conditions. It is essential to include statistical characteristics of the actual speed data (e.g., average, standard deviation, or the 95% confidence interval) when individuals are traversing different types of terrain, derived from real datasets. Additionally, incorporating statistical characteristics of the ground truth data in Figure 4 would strengthen the paper's credibility and the evaluation of the model's performance.

A new table (Table 1) and figure (Figure 4) have been added which illustrate the breakdown of the tracks by slope angle, and the confidence intervals for the mean walking speed. These demonstrate how our model closely fits the average walking speed, particularly in areas where the majority of walking occurs (where we have the most data), and thus where accuracy is most important. We have also added an explanation for the deviation from the confidence interval at high slopes [lines 260-272]

Transparency in Exclusion of Data Points: It would be beneficial for readers to know the number and percentage of data points or sections that were excluded due to the filtering process. This transparency will provide a clearer picture of the data selection and processing steps.

Responding to this point is tricky as “data points” were not truly defined or counted until after the initial filtering and merging. This means that there are no equivalent numbers for data points present in the unfiltered data that would support a direct comparison. However, we support the need for transparency and therefore all code used to filter the data is included for checking. The initial number of GPS tracks (~20,000) has been included in both Supplementary Information 2, and in the results. This reduced to ~7600 after processing (also mentioned in the results). The majority of the removed data was from non-walking tracks (mostly clear from the velocity), but a detailed breakdown of exclusions by filtering reasons was not recorded. The main details of filtering are in Supplementary Information 2 but the results section has been changed to make the need for significant filtering clearer. [Lines 217-220]

Additional Minor Comments:

Introduction References: In the introduction section, consider providing more references when discussing the classification of factors that can impact walking speed into two groups. This will enhance the depth of the literature review and provide a stronger foundation for your work.

Further references have been added, which discuss the factors which can affect walking speeds. [Line 15]

Model Performance Evaluation: Highlighting the fact that the model's performance evaluations are conducted at the section level should be done earlier in the paper to prevent any misunderstanding. Additionally, consider presenting statistical characteristics of the sections, such as their length and the distribution of slopes, to provide a comprehensive assessment.

A line has been added to the start of results, with the minimum and average point distance. [lines 222/223]. Table 1 has been added detailing the slope distributions of the data

Clarify Terminology: Distinguish between "hill slope" and "walking slope" throughout the paper to avoid confusion. For instance, in the caption for Figure 3, specify the meaning of "slope" in Figures 3A and 3B to enhance clarity for readers.

Figure captions have been adjusted as requested.

---

## [Decision Letter · Decision Letter 2]

1 Dec 2023

Improved prediction of hiking speeds using a data driven approach

PONE-D-23-19311R2

Dear Dr. Wood,

We’re pleased to inform you that your manuscript has been judged scientifically suitable for publication and will be formally accepted for publication once it meets all outstanding technical requirements.

Kind regards,

Yuxia Wang

Academic Editor

PLOS ONE

Additional Editor Comments (optional):

Reviewers' comments:

Reviewer's Responses to Questions

**Comments to the Author**

1. If the authors have adequately addressed your comments raised in a previous round of review and you feel that this manuscript is now acceptable for publication, you may indicate that here to bypass the “Comments to the Author” section, enter your conflict of interest statement in the “Confidential to Editor” section, and submit your "Accept" recommendation.

Reviewer #1: All comments have been addressed

2. Is the manuscript technically sound, and do the data support the conclusions?

Reviewer #1: Yes

3. Has the statistical analysis been performed appropriately and rigorously? 

Reviewer #1: Yes

4. Have the authors made all data underlying the findings in their manuscript fully available?

Reviewer #1: Yes

5. Is the manuscript presented in an intelligible fashion and written in standard English?

Reviewer #1: Yes

6. Review Comments to the Author

Reviewer #1: I thank the authors for addressing my comments thoroughly and am happy to recommend that this revised version be accepted for publication.

7. PLOS authors have the option to publish the peer review history of their article (what does this mean?). If published, this will include your full peer review and any attached files.

Reviewer #1: No

---

## [Editor Report · Acceptance letter]

8 Dec 2023

PONE-D-23-19311R2 

Improved prediction of hiking speeds using a data driven approach 

Dear Dr. Wood:

I'm pleased to inform you that your manuscript has been deemed suitable for publication in PLOS ONE. Congratulations! Your manuscript is now with our production department. 

Kind regards, 

on behalf of

Dr. Yuxia Wang 

Academic Editor

PLOS ONE